Lp-PLA2 silencing ameliorates inflammation and autophagy in nonalcoholic steatohepatitis through inhibiting the JAK2/STAT3 pathway

Yao Jinmei
Zhao Ying Yingzhao@zju.edu.cn
Department of Clinical Laboratory, Key Laboratory of Clinical In Vitro Diagnostic Techniques of Zhejiang Province, The First Affiliated Hospital, Zhejiang University, School of Medicine , Hangzhou, Zhejiang Province , China
Dong Peixin
Electronic publication date: 2023 Jun 26
Publication date: 2023
Volume: 11
Electronic Location ID: e15639
Received 2023 Mar 22; Accepted 2023 Jun 5
Copyright: © 2023 Yao and Zhao
Copyright year: 2023
Copyright holder: Yao and Zhao
License: This is an open access article distributed under the terms of the Creative Commons Attribution License, which permits unrestricted use, distribution, reproduction and adaptation in any medium and for any purpose provided that it is properly attributed. For attribution, the original author(s), title, publication source (PeerJ) and either DOI or URL of the article must be cited.
License URL: https://creativecommons.org/licenses/by/4.0/

Keywords: Nonalcoholic steatohepatitis, Lp-PLA2, JAK2/STAT3 pathway, Autophagy, Kupffer cells

Funding: National Natural Science Foundation of Zhejiang Province LY19H200003 National Natural Science Foundation of Zhejiang Province LY20H200005 This work was supported by the National Natural Science Foundation of Zhejiang Province (No. LY19H200003) and the National Natural Science Foundation of Zhejiang Province (No. LY20H200005). The funders had no role in study design, data collection and analysis, decision to publish, or preparation of the manuscript.

==============================
Background

Nonalcoholic steatohepatitis (NASH), a common cause of liver-related morbidity and mortality worldwide, is characterized by inflammation and hepatocellular injury. Our research focuses on lipoprotein-associated phospholipase A2 (Lp-PLA2), an inflammation-related biomarker that has recently garnered interest in the context of NASH due to its potential roles in disease pathogenesis and progression.

Methods

We established a NASH mouse model using a high-fat diet (HFD) and treated it with sh-Lp-PLA2 and/or rapamycin (an mTOR inhibitor). Lp-PLA2 expression in NASH mice was detected by qRT-PCR. Serum levels of liver function parameters and inflammatory cytokines were detected using corresponding assay kits. We examined pathological changes in liver using hematoxylin-eosin, oil red O, and Masson staining, and observed autophagy through transmission electron microscopy. The protein levels of Lp-PLA2, mTOR, light chain 3 (LC3) II/I, phosphorylated Janus kinase 2 (p-JAK2)/JAK2, and phosphorylated signal transducer and activator of transcription 3 (p-STAT3)/STAT3 were determined by western blotting. Kupffer cells extracted from C57BL/6J mice were treated to replicate NASH conditions and treated with sh-Lp-PLA2, rapamycin, and/or a JAK2-inhibitor to further verify the roles and mechanisms of Lp-PLA2 in NASH.

Results

Our data indicate an upregulation of Lp-PLA2 expression in HFD-induced NASH mice. Silencing Lp-PLA2 in NASH mice reduced liver damage and inflammation markers (aspartate aminotransferase (AST), alanine aminotransferase (ALT), total cholesterol (TC), triglycerides (TG), tumor necrosis factor-alpha (TNF-α), and interleukin-6 (IL-6)), while increasing IL-10 levels, an anti-inflammatory cytokine. Additionally, Lp-PLA2 silencing decreased lipid and collagen accumulation and promoted autophagy. The beneficial effects of sh-Lp-PLA2 on NASH were enhanced by rapamycin. Furthermore, Lp-PLA2 silencing resulted in the downregulation of the expression of p-JAK2/JAK2 and p-STAT3/STAT3 in NASH mice. Similar results were observed in Kupffer cells treated under NASH conditions; Lp-PLA2 silencing promoted autophagy and repressed inflammation, effects which were potentiated by the addition of rapamycin or a JAK2-inhibitor.

Conclusion

Our findings suggest that silencing Lp-PLA2 promotes autophagy via deactivating the JAK2/STAT3 signaling pathway, thereby restraining NASH progression. This highlights the potential therapeutic value of targeting Lp-PLA2, adding a new dimension to our understanding of NASH pathogenesis and treatment strategies.

Introduction

Nonalcoholic fatty liver disease (NAFLD) is a type of chronic liver disease, ranging from hepatic steatosis to non-alcoholic steatohepatitis (NASH) and finally to cirrhosis (Estes et al., 2018; Manne, Handa & Kowdley, 2018). The incidence of NAFLD in Chinese adult is 15–30%, of which, 20–30% patients with NAFLD progress to NASH (Hu & Zhang, 2016). Meanwhile, 10–15% of patients with NASH eventually progress to end-stage liver disease or even hepatocellular carcinoma (Zhou et al., 2019). Although there are numerous studies on effective therapies for NASH (Younossi et al., 2018; Friedman et al., 2018), its therapeutic strategies in clinical application are still scarce (Xu et al., 2022; Chalasani et al., 2018). This is primarily due to the need for further research, validation in larger-scale trials, assessment of long-term safety, and addressing challenges related to patient adherence. Examples include antifibrotic drugs that have shown promise in reducing NASH-related liver fibrosis and inflammation (Stefan, Häring & Cusi, 2019), but require additional research to evaluate long-term efficacy and safety. Additionally, targeting specific molecular pathways like the peroxisome proliferator-activated receptor alpha (PPAR-α) pathway has shown potential but requires validation in larger clinical trials (Francque et al., 2021).

Lipoprotein-associated phospholipase A2 (Lp-PLA2) is a calcium-independent enzyme (Xu et al., 2020). In blood circulation, Lp-PLA2 regulates vascular inflammation in lipid metabolism, therefore, it plays an essential role in vascular inflammation-related diseases (Huang, Wang & Shen, 2020). Colak et al. (2012) suggested that Lp-PLA2 expression is upregulated in NAFLD including NASH, which is closely associated with histological steatosis scores in patients. Sun et al. (2017) revealed that the reduction of macrophage Lp-PLA2 can mitigate the inflammatory cell infiltration in liver tissues from mice. However, the role of Lp-PLA2 in NASH remains to be investigated.

In patients with NASH, autophagy was found to be impaired in liver tissues (Park et al., 2021; González-Rodríguez et al., 2014; Zeng et al., 2015). JAK2/STAT3 pathway is involved in regulating multiple physiological and pathological processes, including inflammation, oxidative stress, and autophagy (An et al., 2021). Research showed that the suppression of the JAK2/STAT3/VEGFA pathway induces the protective autophagy and apoptosis in non-small cell lung cancer cells (Liang et al., 2019). Xu et al. (2021) indicated that suppressing the JAK2/STAT3/SOCS3 pathway can ameliorate liver fibrosis and inflammatory response. Attia et al. (2021) demonstrated that the central role of the SOCS3/JAK2/STAT3 pathway in the development and progression of NASH to hepatocellular carcinoma. However, it has not been reported whether Lp-PLA2 regulates inflammation and autophagy in NASH via the JAK2/STAT3 pathway.

In the context of the current understanding of NASH and the potential role of Lp-PLA2, significant knowledge gaps remain. Specifically, the precise regulatory effects of Lp-PLA2 on liver function, inflammation, and autophagy in NASH have not been fully elucidated. Furthermore, the potential mechanisms through which Lp-PLA2 might interact with the JAK2/STAT3 signaling pathway in the context of NASH remain unexplored. The present study, therefore, aimed to fill these gaps by establishing a mouse model and a Kupffer cell model of NASH. Through these models, we sought to comprehensively investigate the role and mechanistic implications of Lp-PLA2 in the context of NASH, with a particular focus on its interaction with the JAK2/STAT3 signaling pathway.

Materials and Methods

Animal experiments

The protocol on animal was approved by the Institutional Animal Care and Use Committee of The First Affiliated Hospital, Zhejiang University, School of Medicine (No. 965-2019) and conformed to the provisions of the Declaration of Helsinki. A total of 36 male C57BL/6J mice aged 4 weeks were purchased from Cavens Lab Animal Co., Ltd. (Changzhou, China). All mice were raised in a pathogen-free condition at 23 ± 3 °C and 55 ± 10% humidity with a 12-h light-dark cycle in a clean-conventional animal room, with ad libitum access to food and water. All mice were stochastically divided into six groups (n = 6): normal caloric diet (NCD, control) group, high fat diet (HFD, model) group, sh-negative control (sh-NC) + HFD group, sh-Lp-PLA2 + HFD group, rapamycin + HFD group, and sh-Lp-PLA2 + rapamycin + HFD group. Mice in the NCD group was fed with NCD as control, and those in the HFD group were given HFD to induce NASH. The HFD used in this study contained 60% kcal from fat, 20% kcal from carbohydrates, and 20% kcal from protein (Research Diets, NJ, USA). Mice in the sh-NC + HFD and sh-Lp-PLA2 + HFD groups were respectively injected with lentivirus containing sh-NC and sh-Lp-PLA2 by tail vein, and fed with HFD. Mice in the rapamycin + HFD group were injected intraperitoneally with 1 mg/kg/day rapamycin (mTOR inhibitor; Sigma-Aldrich, St. Louis, MO, USA) and fed with HFD. Mice in the sh-Lp-PLA2 + rapamycin + HFD group were treated with sh-Lp-PLA2, rapamycin, and HFD together. All mice were weighed once a week for 12 weeks. Then, serum samples were obtained from mice, and liver and adipose tissues were collected after euthanized by gradual-fill CO2 asphyxiation and cervical dislocation.

qRT-PCR

Total RNAs were isolated from the adipose and liver tissues from mice using the Trizol kit (Invitrogen, Waltham, MA, USA). RNA weas reversely transcribed to cDNA by the Reverse Transcription Kit (Fermentas, Waltham, MA, USA). qRT-PCR was carried out using the SYBR® Premix Ex TaqTM II Kit (Action-award Biological Technology Co., Ltd., Guangzhou, Guangdong, China) and the Mx3000P qPCR system (Agilent, Santa Clara, CA, USA). The reaction condition was 95 °C for 3 min and then 40 cycles of 95 °C for 12 s and 62 °C for 40 s. Primer sequences were Lp-PLA2-F (5′-TCA CAA GAC TCC AAT CGG TCA G-3′) and Lp-PLA2-R (5′-CGA CGG GGT ACG ATC CAT TTC-3′); GAPDH-F (5′-GAA GGT CGG TGT GAA CGG ATT TG-3′) and GAPDH-R (5′-CAT GTA GAC CAT GTA GTT GAG GTC A-3′). GAPDH is the internal reference. Relative gene expression was obtained by the 2−∆∆Ct method.

Hematoxylin and eosin (H&E) staining

Embedded liver tissues were sectioned into a thickness of 4 µm. Tissue sections were stained with hematoxylin and eosin (Keygen Biotech, Nanjing, China), followed by observation under an optics microscope (Olympus, Tokyo, Japan).

Oil red O staining

Liver tissues from mice were sectioned to a thickness of 6 µm using a freezing microtome. Sections were subjected to Oil red O staining according to a preceding literature (Kozlova et al., 2022).

Masson staining

Liver tissues from mice were fixed with 4% paraformaldehyde and embedded in paraffin for sectioning into 4 µm thickness. Tissue sections were stained with hematoxylin for 5 min and differentiated with 0.1% HCl, followed by staining with ponceau red liquid dye acid complex for 5 min. Then, sections were incubated with 1% phosphomolybdic acid solution for 50 s and with methyl green solution for 5 s. Finally, sections were observed under an optical microscope (Olympus, Tokyo, Japan).

Transmission electron microscopy

Autophagy in liver tissues was determined by TEM. Briefly, liver tissues were immobilized with 2.5% glutaraldehyde and sectioned into 1 mm3. Then, sections underwent dehydration, embedding, solidification, and standard uranyl acetate staining. TEM images were acquired with the Transmission Electron Microscope HF5000 (Hitachi, Chiyoda-Ku, Tokyo, Japan).

Isolation of Kupffer cells

Kupffer cells were separated from C57BL/6 mice according to a reference by Li et al. (2021). In brief, livers of mice were perfused with 10 mL calcium-free Hank balanced salt solution (HBSS; HyClone Laboratories, San Angelo, TX, USA) via the portal vein, and then with 0.27% type IV collagenase (Sigma-Aldrich, St. Louis, MO, USA) in a water bath for 20 min at 37 °C. Subsequently, perfused livers were dissected, and single-cell suspension was prepared using a 70 mm cell filter. Hepatocytes and nonparenchymal cells were separated from cell suspension by centrifuging at 50×g and 4 °C for 3 min. Nonparenchymal cell-containing solution was suspended with HBSS, gently overlaid onto a two-step Percoll gradient (Sigma-Aldrich, St. Louis, MO, USA), and then centrifuged at 1,800×g and 4 °C for 15 min without break. Cells in the middle layer (Kupffer cells) were collected and cultured into Dulbecco’s Modified Eagle Medium (DMEM) with 10% fetal bovine serum (FBS; Thermo Fisher Scientific, Waltham, MA, USA), 100 U/mL penicillin/streptomycin, and 2 mM L-glutamine at 37 °C with 5% CO2. After incubation for 2 h, the isolated Kupffer cells were purified by removing the non-adherent cells.

Kupffer cell treatment

After 24 h culture of isolated Kupffer cells, palmitic acid (PA) and lipopolysaccharide (LPS) were treated to cells for establishing NASH cell model. Kupffer cells with NASH were transfected with lentivirus containing sh-NC or sh-Lp-PLA2, and/or treated with rapamycin or JAK2 inhibitor (Tocris Bioscience, Bristonl, UK).

Monodansylcadaverine (MDC) staining

MDC was applied as a specific fluorescent marker for autophagosome formation. Kupffer cells were cultivated in 24-well plates for 12 h at 37 °C with 5% CO2. Cells were incubated with 0.05 mM MDC probe (Solarbio, Beijing, China) in PBS for 30 min away from light. Fluorescent images were obtained by a confocal laser scanning microscope (Olympus, Tokyo, Japan).

Western blot assay

Liver tissues and Kupffer cells were lysed using the RIPA lysis buffer (TaKaRa, Kusatsu, Shiga, Japan) for extracting total protein. Proteins were subjected to quantification and western blotting as previously reported protocol (Wu et al., 2020). Primary antibodies for Lp-PLA2, mTOR, p-mTOR, LC3II/I, JAK2, phosphor (p)-JAK2, STAT3, p-STAT3, and GAPDH (Abcam, Cambridge, UK) were applied in this study. Protein bands were developed using a West Chemiluminescent Substrate (Thermo Fisher Scientific, Waltham, MA, USA) and observed using the Bio-Rad CD Touch detection system (Bio-Rad, Hercules, CA, USA).

Flow cytometry

Kupffer cells (1 × 105 cells/mL) were mixed with 100 µL FACS stain buffer (BD Bioscience, San Jose, CA, USA), followed by staining with CD11c (M1 macrophage marker) and CD206 (M2 macrophage marker) for 30 min under darkness to analyze Kupffer cell polarization. Samples were resuspended with the stain buffer, and then were analyzed using the FlowJo software package (Tree Star, Ashland, OR, USA).

Assessment of biochemical markers and cytokines

The following biochemical parameters included alanine aminotransferase (ALT), aspartate aminotransferase (AST), total cholesterol (TC), and triglycerides (TG) were measured in serum level by a Roche C311 automatic biochemistry analyzer using assay-specific Roche reagents (Hitachi Corp., Ibaragi, Japan). The pro-inflammatory cytokines (TNF-α and IL-6), anti-inflammatory cytokines (IL-10), and arginase-1 (Arg-1) were detected by enzyme-linked immunosorbent assay (ELISA) and Mlbio reagents (Shanghai Mlbio, Shanghai, China).

Statistical analyses

All data were expressed as mean ± standard deviation. Differences between two groups were analyzed using Student’s t-test. For comparisons among three or more groups, one-way analysis of variance (ANOVA) was used. A P-value of less than 0.05 was considered statistically significant, and all statistical tests were two-sided. The statistical analysis was conducted by GraphPad version 7.0 software.

Results

Lp-PLA2 silencing ameliorates liver damage and inflammation in HFD mice probably via affecting autophagy

To explore the role of Lp-PLA2 in NASH, the NASH mouse model was established by feeding with HFD and then transfected with sh-Lp-PLA2. Lp-PLA2 expression was higher in the adipose and hepatic tissues of HFD mice compared to control (NCD) mice, and Lp-PLA2 knockdown reduced its expression (P < 0.05; Fig. 1A). The body weight and liver weight/index were increased in HFD mice, which were rescued by Lp-PLA2 knockdown (P < 0.001; Figs. 1B and 1C), indicating a potential role for Lp-PLA2 in weight regulation. Moreover, AST and ALT are two plasma enzymes that play the essential role in maintaining liver integrity (Miguel et al., 2022). TC and TG are important indicators of lipid accumulation (Ma et al., 2022). Figure 1D shows that AST, ALT, TC, and TG levels in serum of HFD mice presented a significant increase in comparison with that in NCD mice, while Lp-PLA2 knockdown reduced the levels of these biomarkers in HFD mice (P < 0.01), suggesting an improvement in liver function and lipid metabolism. Furthermore, inflammation is a critical pathological feature in NASH, reflected by the pro-inflammatory (TNF-α and IL-6) and anti-inflammatory cytokines (IL-10). A significant increase of TNF-α and IL-6 was observed in serum of HFD mice when compared with that in NCD mice, conversely, IL-10 presented a decreased trend (P < 0.001). However, Lp-PLA2 knockdown ameliorated the inflammatory response of HFD mice, as evidenced by the decreased TNF-α and IL-6 levels, and the increased IL-10 level compared to the HFD group (P < 0.05; Fig. 1E).

Figure 1 Silencing of Lp-PLA2 and promoting of autophagy alleviates liver injury and inflammation in high-fat diet (HFD)-induced nonalcoholic steatohepatitis (NASH) mice.

Mice were fed with HFD and treated with sh-Lp-PLA2 and/or rapamycin. (A) The expression of Lp-PLA2. (B) The body weight of mice. (C) The liver weight and liver index of mice. (D) The serum levels of liver function parameters (AST, ALT, TC, and TG). (E) The serum levels of pro-inflammatory cytokines (TNF-α and IL-6) and anti-inflammatory cytokines (IL-10). *P < 0.05, **P < 0.01, ***P < 0.001.

In addition, autophagy is an essential mechanistic pathway in NASH, which is involved in liver injury and immune responses (Amir & Czaja, 2011). mTOR regulates cell growth and autophagy, here, rapamycin (a mTOR inhibitor) was applied to treat HFD mice for assessing the role of autophagy during Lp-PLA2 regulating NASH. The body weight of HFD mice was significantly reduced after rapamycin treatment (Fig. 1B). Similarly, the weight and index of the liver were decreased in HFD mice treated with rapamycin, indicating an improvement in liver pathology (P < 0.001; Fig. 1C). Moreover, the serum levels of AST, ALT, TC, TG, TNF-α, and IL-6 were significantly reduced in HFD mice treated with rapamycin, and IL-10 level was increased (P < 0.05; Figs. 1D and 1E). Of note, rapamycin treatment enhanced the effects of Lp-PLA2 knockdown on inhibiting liver injury and inflammation (P < 0.05; Figs. 1C–1E). These findings suggest that Lp-PLA2 silencing ameliorates liver damage and inflammation in HFD mice, potentially through its effects on autophagy.

Lp-PLA2 silencing alleviates HFD-induced hepatological injury probably via affecting autophagy

Our morphological observations of liver tissues in HFD mice revealed several pathophysiological changes. This included diffuse steatosis, ballooning degeneration, and obvious inflammatory cell infiltration in the lobules and portal areas, changes that were significantly more marked than in mice on a NCD (Machado et al., 2015; Asgharpour et al., 2016). These observations confirmed the successful establishment of the NASH model. Lp-PLA2 knockdown or rapamycin treatment ameliorated the hepatological injury in HFD mice, particularly when Lp-PLA2 knockdown combined with rapamycin administration (Fig. 2A). The mitigated hepatological injury suggests a potential therapeutic role for Lp-PLA2 silencing and rapamycin in NASH. The accumulation of lipids within hepatocytes, a key pathological feature of NASH, was notably more abundant in the liver tissues of HFD mice as evidenced by Oil Red O staining. However, we found that Lp-PLA2 silencing or rapamycin treatment significantly decreased this lipid accumulation (Fig. 2B). This suggests that these interventions may have a role in reversing the lipid accumulation associated with NASH. In addition to the lipid accumulation, hepatic fibrosis is another major complication of NASH. We applied Masson’s staining to the liver tissues to assess the effects of Lp-PLA2 silencing and autophagy, induced by rapamycin, on hepatic fibrosis. The findings revealed that Lp-PLA2 knockdown or rapamycin treatment notably mitigated the HFD-induced collagen accumulation, a surrogate marker for fibrosis, in the mice (Fig. 2C). Importantly, we found a synergistic effect when combining Lp-PLA2 silencing and rapamycin treatment. This combination was more effective at inhibiting both lipid and collagen accumulation in the liver tissues of HFD mice than either treatment alone (Figs. 2B and 2C). This suggests that a combined treatment strategy may be more beneficial in the management of NASH.

Figure 2 Silencing of Lp-PLA2 and promoting of autophagy alleviates liver lipid and collagen accumulation in HFD-induced NASH mice.

Mice were fed with HFD and treated with sh-Lp-PLA2 and/or rapamycin. (A–C), Representative histology of hematoxylin-eosin (HE), Oil red O, and Masson staining (scale bar = 50 µm).

Lp-PLA2 silencing promotes autophagy and the JAK2/STAT3 pathway in liver architecture of HFD-induced NASH mice

The evaluation of the TEM micrographs of liver sections was conducted to validate the regulatory role of Lp-PLA2 on autophagy in NASH. TEM analysis showed the increased autophagosomes in HFD mice compared to NCD mice, indicating an induction of autophagy in response to an HFD. Both Lp-PLA2 knockdown and rapamycin treatment, however, reduced the quantity of these autophagosomes in HFD mice. Interestingly, rapamycin was found to amplify the repressive effect of Lp-PLA2 knockdown on autophagy (Fig. 3A). We also assessed the LC3-II/I ratio, a widely accepted marker of autophagy, which showed a decrease in HFD mice compared to NCD mice (P < 0.001). This decrease suggests a reduced level of autophagy in the HFD group. However, either Lp-PLA2 knockdown or rapamycin treatment, or their combination, was effective in increasing LC3-II/I expression in HFD mice (P < 0.05; Fig. 3B). This further supports the inhibitory role of Lp-PLA2 knockdown and rapamycin treatment on autophagy in NASH.

Figure 3 Lp-PLA2 silencing promotes autophagy and inhibits the JAK2/STAT3 pathway in HFD-induced NASH mice.

Mice were fed with HFD and treated with sh-Lp-PLA2 and/or rapamycin. (A) The autophagosomes in liver tissues of mice (scale bar = 1 µm). (B) The protein levels of Lp-PLA2, p-mTOR/mTOR, LC3II/I, p-JAK2/JAK2, and p-STAT3/STAT3. *P < 0.05, **P < 0.01, ***P < 0.001.

On the molecular level, we examined the JAK2/STAT3 signaling pathway, known to be a critical regulator in NASH and to be closely associated with autophagy (An et al., 2021). Western blotting presented that the expression of Lp-PLA2, p-mTOR/mTOR, p-JAK2/JAK2, and p-STAT3/STAT3 all were remarkably increased in HFD mice; however, Lp-PLA2 knockdown was reversed in expression (P < 0.05; Fig. 3B). These findings suggest that Lp-PLA2 may regulate autophagy in NASH via the JAK2/STAT3 signaling pathway.

Lp-PLA2 silencing promotes NASH-induced Kupffer cell autophagy via suppressing JAK2/STAT3 pathway

In order to further verify the suppressive effect of Lp-PLA2 silencing on NASH-induced autophagy, we turned our focus to Kupffer cells, the resident macrophages in the liver, extracted from mice. MDC, a fluorescent dye, is applied for the measurement of autophagic vacuoles in Kupffer cells. We noticed a relatively weak MDC fluorescence intensity in Kupffer cells from the NASH group compared to the normal group, indicating a lower level of autophagic vacuoles (Fig. 4A). When the Kupffer cells from the NASH group were treated with sh-Lp-PLA2, rapamycin, or JAK2-inhibitor, we observed enhanced punctate MDC fluorescence intensity, suggesting increased autophagic activity. Of note, rapamycin or JAK2-inhibitor addition enhanced the promotive effect of sh-Lp-PLA2 on autophagy in Kupffer cells with NASH (Fig. 4A). We further evaluated the LC3-II/I ratio and found that it was lower in Kupffer cells from the NASH group compared to normal cells. This reduction was effectively reversed by treatment with sh-Lp-PLA2, rapamycin, and/or JAK2-inhibitor (P < 0.01), indicating a restored level of autophagy. Parallel to our findings in the HFD-induced NASH mice, we observed a significant increase in the protein expression of Lp-PLA2, p-mTOR/mTOR, p-JAK2/JAK2, and p-STAT3/STAT3 in Kupffer cells from the NASH group compared to normal cells (P < 0.001). Interestingly, upon Lp-PLA2 knockdown, these increases were diminished, further implicating the JAK2/STAT3 pathway in the regulation of autophagy by Lp-PLA2 (P < 0.05; Fig. 4B).

Figure 4 Lp-PLA2 silencing promotes autophagy via deactivating the JAK2/STAT3 pathway in Kupffer cells induced with NASH.

Kupffer cells were isolated from mice and treated with sh-Lp-PLA2, rapamycin, and/or JAK2-inhibitor. (A) Autophagy was detected by monodansylcadaverine (MDC) staining (scale bar = 25 µm). (B) The protein levels of Lp-PLA2, p-mTOR/mTOR, LC3II/I, p-JAK2/JAK2, and p-STAT3/STAT3. *P < 0.05, **P < 0.01, ***P < 0.001.

Lp-PLA2 silencing inhibits NASH-induced the M1 polarization of Kupffer cells via enhancing the JAK2/STAT3 pathway-mediated autophagy

Previous studies have reported that macrophage polarization acts as an essential role in NASH (Yao et al., 2020). As part of this polarization process, macrophages may shift towards either a pro-inflammatory (M1) or an anti-inflammatory (M2) phenotype. CD11c and CD206 are M1 and M2 macrophage marker proteins, respectively. In our study, we observed a notably higher ratio of CD11c+/CD206− (M1 phenotype) in Kupffer cells in the NASH group compared to normal cells, indicating a shift towards a pro-inflammatory state (P < 0.001; Fig. 5A). However, the increase in CD11c+/CD206− ratio in NASH group was suppressed by sh-Lp-PLA2, rapamycin, and/or JAK2-inhibitor treatment (P < 0.01). On the contrary, the CD206+/CD11c− ratio (M2 phenotype) presented opposing results to the CD11c+/CD206− ratio across different groups, indicating a more anti-inflammatory state (Fig. 5A).

Figure 5 Lp-PLA2 silencing inhibits M1 polarization of Kupffer cells induced with NASH.

Kupffer cells were isolated from mice and treated with sh-Lp-PLA2, rapamycin, and/or JAK2-inhibitor. (A) M1 (CD11c+/CD206−) and M2 (CD206+/CD11c−) parameters were detected by flow cytometry. (B, C) The levels of pro-inflammatory cytokines (TNF-α and IL-6) and anti-inflammatory cytokines (Arg-1 and IL-10). *P < 0.05, **P < 0.01, ***P < 0.001.

In addition, M1 macrophage activation can exacerbate the inflammatory responses via secreting TNF-α and IL-6, while M2 macrophages can suppress inflammatory by secreting anti-inflammatory cytokines (Arg-1 and IL-10). In line with these characteristics, we found that levels of TNF-α and IL-6 were elevated in NASH-induced Kupffer cells compared to normal cells, as shown by ELISA assays. This pro-inflammatory state was downregulated by treatment with sh-Lp-PLA2, rapamycin, or a JAK2-inhibitor (P < 0.05). Notably, rapamycin or JAK2-inhibitor addition potentiated the inhibitory effect of sh-Lp-PLA2 on TNF-α and IL-6 secretion in Kupffer cells induced with NASH (P < 0.001; Fig. 5B). Simultaneously, Arg-1 and IL-10 levels presented opposite changes with TNF-α and IL-6 levels in different treatment groups (P < 0.05; Fig. 5C).

Discussion

Evidence has showed that impaired autophagy and inflammatory response contribute to NASH (Park et al., 2021). Lp-PLA2 is a biomarker released by immune cells specifically to cause inflammation in blood vessels (Lv et al., 2021). This study showed that Lp-PLA2 silencing can ameliorate liver functions and inflammatory response, and recover the impaired autophagy induced by NASH in mice and Kupffer cells (the resident liver macrophages). Meanwhile, we found that the potential mechanism of Lp-PLA2 silencing against NASH is involved in the deactivation of the JAK2/STAT3 signaling pathway.

Lp-PLA2, a unique enzyme secreted by macrophages, circulates in blood to mediate vascular inflammation and oxidative stress (Huang, Wang & Shen, 2020). Although previous studies have identified the regulatory role of Lp-PLA2 in NAFLD (Colak et al., 2012), its specific role and underlying mechanisms in NASH remain largely unknown. In the current study, an HFD-induced mouse model was constructed to explore the role of Lp-PLA2 in NASH. Here, we found that Lp-PLA2 presented a higher level in HFD-induced NASH mice than that in normal mice; thus, sh-Lp-PLA2 was transfected into HFD-induced NASH mice to further decipher its functions. Our results showed that Lp-PLA2 silencing reduced the increased levels of liver function parameters (AST, ALT, TC, and TG) in HFD-induced NASH mice. Simultaneously, Lp-PLA2 silencing reduced the levels of TNF-α and IL-6 (pro-inflammatory cytokines), and elevated the IL-10 (anti-inflammatory cytokine) level in serum of HFD-induced NASH mice. Lp-PLA2 downregulation showed the consistent anti-inflammatory effect in NASH with that in cardiovascular disease (Sun et al., 2017). In accordance with our findings, a study by Chen et al. (2023) demonstrated that Lp-PLA2 knockdown rabbits had reduced macrophages and inflammatory cytokine expression, and that macrophages of homozygous Lp-PLA2 knockout rabbits were insensitive to M1 polarization. In addition, the unbalance of lipid homeostasis in liver is an important pathological feature in NASH (Ipsen, Lykkesfeldt & Tveden-Nyborg, 2018). Lipid droplet will be accumulated in hepatocytes when NASH happens (Lan et al., 2021), which was also observed in HFD-induced NASH mice in our study. Lp-PLA2 silencing decreased the lipid accumulation in liver tissues of HFD-induced NASH mice. Moreover, NASH is presented as fatty liver with hepatic injury, inflammation, and fibrosis (Cusi, 2012; Sheka et al., 2020). Liver fibrosis is characterized by the excess accumulation of extracellular matrix proteins, covering collagen (Bataller & Brenner, 2005). Masson staining showed that the excessive collagen accumulation in liver tissues in HFD-induced NASH mice, however, which was recovered by Lp-PLA2 silencing. Furthermore, impaired autophagy is a key pathological feature of NASH (Park et al., 2021). According to TEM observation, autophagosomes was reduced in liver tissues of HFD-induced NASH mice, while Lp-PLA2 silencing rescued the number of autophagosomes. Meanwhile, the decreased level of LC3II/I (autophagy biomarker) induced by NASH in mice was upregulated by Lp-PLA2 silencing. Of note, we found that rapamycin (a mTOR inhibitor for promoting autophagy) treatment presented similar effects as Lp-PLA2 silencing on alleviating liver injury, inflammation, lipid and collagen accumulation in HFD-induced NASH mice, additionally, rapamycin markedly enhanced the effects of Lp-PLA2 silencing. These discoveries indicate that Lp-PLA2 silencing protects against NASH via promoting autophagy.

Further, we investigated the molecular mechanisms behind the protective effect of Lp-PLA2 in NASH. JAK2/STAT3 is important intracellular signal transduction pathways involved in inflammatory response and autophagy (Wu et al., 2018; Li et al., 2021). JAK2/STAT3 signal reportedly plays an essential role in the treatment of NASH (Wei et al., 2021). Bi et al. (2018) reported that the inhibition of JAK2/STAT3 pathway suppressed inflammation and hepatocyte steatosis in Kupffer cells (resident liver macrophages) with NAFLD. Chen & Liu (2018) found that actein ameliorated hepatic steatosis and fibrosis in HFD-induced NAFLD via inhibiting JAK2/STAT3 signal. Zhang et al. (2006) discovered that rapamycin suppressed JAK2/STAT3 pathway to reduce inflammation induced by acute liver injury. However, the roles of JAK2/STAT3 pathway in Lp-PLA2 regulating NASH have not been evaluated. In the current study, we observed that the JAK2/STAT3 signaling pathway, which has been implicated in the regulation of inflammation and autophagy, is deactivated by Lp-PLA2 silencing. Our findings are consistent with previous studies showing the therapeutic potential of targeting the JAK2/STAT3 pathway in NASH. We concluded that Lp-PLA2 silencing can deactivate the JAK2/STAT3 pathway, thereby promoting autophagy and repressing inflammation in NASH.

Conclusions

In conclusion, Lp-PLA2 silencing exerts protective effects on NASH via suppressing liver injury, inflammation, and fibrosis. The mechanism of Lp-PLA2 silencing against NASH is involved in the promotion of autophagy induced by JAK2/STAT3 inhibition. These findings highlight the potential of targeting Lp-PLA2 as a novel therapeutic strategy for the treatment of NASH. By developing new inhibitors specifically targeting Lp-PLA2, we can potentially advance the field and provide alternative approaches for managing NASH. The novelty of our findings and their implications suggest promising avenues for further research and clinical translation.

Supplemental Information

Supplemental Information 1 Raw data.

Click here for additional data file.

Supplemental Information 2 ARRIVE List.

Click here for additional data file.

Supplemental Information 3 Original images of Western blot.

Click here for additional data file.

Additional Information and Declarations

Competing Interests

Author Contributions

Animal Ethics

Data Availability

The authors declare that they have no competing interests.

Jinmei Yao conceived and designed the experiments, performed the experiments, analyzed the data, authored or reviewed drafts of the article, and approved the final draft.

Ying Zhao conceived and designed the experiments, performed the experiments, analyzed the data, prepared figures and/or tables, authored or reviewed drafts of the article, and approved the final draft.

The following information was supplied relating to ethical approvals (i.e., approving body and any reference numbers):

The First Affiliated Hospital, College of Medicine, Zhejiang University full approval for this research (No. 965-2019).

The following information was supplied regarding data availability:

The raw measurements are available in the Supplemental File.

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
