# Peer review of "Lp-PLA2 silencing ameliorates inflammation and autophagy in nonalcoholic steatohepatitis through inhibiting the JAK2/STAT3 pathway"

_PeerJ, doi:10.7717/peerj.15639_

## Round 0.1 · original submission · Major Revisions

Please revise your manuscript according to the comments of our Reviewers.

Reviewer 1 ·

Basic reporting

no comment

Experimental design

no comment

Validity of the findings

no comment

Additional comments

Yao J et al. established NASH mouse model and cell model and aim to investigate the role and mechanism of an inflammation-related biomarker Lp-PLA2 in NASH. The authors indicated Lp-PLA2 silencing exerted protective effects on NASH via suppressing liver injury, inflammation, and fibrosis. The mechanism of Lp-PLA2 silencing against NASH was involved in the promotion of autophagy induced by JAK2/STAT3 inhibition. This is a very valuable study. They researched the association between Lp-PLA2 and NASH. But the paper is needed a major revision and the problems are as follows:
1.In page 7 lines 41-42, “Although there are numerous studies on 42 effective therapies for NASH, its therapeutic strategies in clinical application are still scarce.”, Please mark the references.

2.In page 14 lines 182-184, “Similarly, Lp-PLA2 knockdown ameliorated the inflammatory response of HFD mice, with the elevated TNF- αand IL-6 levels, and the reduced IL-10 level (P < 0.05; Figure 1E).”. which group are these results compared with? Compared with NCD group, TNF-α and IL-6 was increased and IL-10 was decreased; compared with HFD group, TNF-α and IL-6 was decreased and IL-10 was increased.

3.Chen J et al. (As follows) found Lp-PLA2 knockdown rabbits were characterized by reduced macrophages and the expression of inflammatory cytokines, and macrophages of homozygous Lp-PLA2 knockout rabbits were insensitive to M1 polarization. This paper studies the association between Lp-PLA2 and M1 polarization in macrophages, and I suggest that the authors cite this paper in the discussion section.
Chen J, Zhang H, Li L, et al. Lp-PLA2 (Lipoprotein-Associated Phospholipase A2) Deficiency Lowers Cholesterol Levels and Protects Against Atherosclerosis in Rabbits. Arterioscler Thromb Vasc Biol. 2023 Jan;43(1):e11-e28. doi: 10.1161/ATVBAHA.122.317898. Epub 2022 Nov 22. PMID: 36412196.

Reviewer 2 ·

Basic reporting

no comment

Experimental design

no comment

Validity of the findings

no comment

Additional comments

The authors concluded that Lp-PLA2 silencing can deactivate the JAK2/STAT3 pathway, thereby promoting autophagy and repressing inflammation in NASH, and rapamycin or JAK2-inhibitor treatment potentiated the promotive effects of Lp-PLA2 silencing on autophagy and M1 polarization of Kupffer cells induced with NASH. This paper is the first to propose the relationship between Lp-PLA2 and autophagy in NASH model. The results of this study have implications for the subsequent development of NASH drugs and large population studies.
However, this study needs to be revised. The main problems are as follows:
1. What is the recipe for a high-fat diet?
2. What is the manufacturer of rapamycin and JAK2 inhibitor?
3. What is the magnification of Figures 2 and 3?
4. Is transmission electron microscopy of liver tissue sections performed by professionals?
5. In pages 14-15 lines 197-199, the author describes: “Morphological observations of liver tissues showed diffuse steatosis, ballooning degeneration and obvious inflammatory cell infiltration in the lobules and portal areas in HFD mice relative to that in NCD mice.”, is this a typical change of liver histomorphology in HFD mice? Does it show that the author established HFD model successfully? Is there any literature to support it?

---

## Round 0.2 · Major Revisions

This revised study presents intriguing findings regarding the role of Lp-PLA2 in NASH and the therapeutic potential of Lp-PLA2 silencing. However, additional work is required to provide a more accurate interpretation of the results, and a more comprehensive discussion and conclusion.

Major issues:
1. First, the Abstract should be improved:
1.1. The abstract could benefit from a sentence or two explaining why the investigation of Lp-PLA2 is important. Why this particular biomarker? How might the results of this study change our understanding of NASH or its treatment?
1.2. There are a lot of abbreviations in the abstract (NASH, Lp-PLA2, HFD, mTOR, AST, ALT, TC, TG, TNF-α, IL-6, IL-10, LC3II/I, p-JAK2/JAK2, p-STAT3/STAT3). Although many of these are standard in the field, it would be good to spell out less common abbreviations upon first usage for clarity and to make the abstract more accessible to a broader audience.
1.3. It's not clear what "Sh-Lp-PLA2 decreased the levels of AST, ALT, TC, TG, TNF-α, and IL-6, and increased IL-10 level in NASH mice" means without context. How do they relate to the progression of NASH?
1.4. The conclusion could be more explicit about the potential clinical implications of the findings. If Lp-PLA2 silencing is effective in controlling NASH progression, does this suggest new therapeutic strategies?
2. Introduction section: The authors also clearly state the objectives of the study. However, there are some points to consider:
2.1. There are several claims made that are not supported by references. For example, the claim about the incidence of NAFLD in Chinese adults and the percentage of NAFLD patients who progress to NASH should be supported by a reference.
2.2. The authors mention that there are numerous studies on effective therapies for NASH but therapeutic strategies in the clinical application are still scarce. It would be helpful to mention a few examples of these therapies and why they haven't made it to clinical application.
3. Results section:
3.1. The authors present many results on the effects of Lp-PLA2 silencing and rapamycin treatment in HFD mice and Kupffer cells but do not provide sufficient explanation or interpretation of these results. Please make sure to explain the results in the text, not just in the figures.
3.2. Ensure that all results have an accompanying figure or table that is referenced in the text.
3.3. Check that all statistical tests performed are appropriate for the data and that the level of significance (P-value) is indicated.

Minor issues:
1. some minor grammar errors could be corrected, such as "Kupffer cells extracted from C57BL/6J mice were induced with NASH" – cells are not typically "induced with" a disease, they are typically "treated with" or "exposed to" a condition that causes the disease. 
2. Belongs? Or belongs to?
3. In which, 20%-30% patients, or of which, 20%-30% patients?
4. Was established to investigated, or were established to investigate?
5. Involving in the JAK2/STAT3 signaling pathway, or involving the JAK2/STAT3 signaling pathway?
6. Liver tissues of mice, should be Liver tissues from mice.
7. Sections went through dehydration, embedding, should be sections underwent dehydration, embedding.
8. An unique, should be a unique.
9. "Further, sh-Lp-PLA2, rapamycin, and JAK2-inhibitor treated Kupffer cells induced with NASH to validate..." is unclear. It could be rephrased as "Further, we treated NASH-induced Kupffer cells with sh-Lp-PLA2, rapamycin, and a JAK2-inhibitor to validate...".

Reviewer 1 ·

Basic reporting

no comment

Experimental design

no comment

Validity of the findings

no comment

Additional comments

no comment

Reviewer 2 ·

Basic reporting

no comment

Experimental design

no comment

Validity of the findings

no comment

Additional comments

no comment

Reviewer 3 ·

Basic reporting

No comment

Experimental design

No comment

Validity of the findings

No comment

Annotated reviews are not available for download in order to protect the identity of reviewers who chose to remain anonymous.

---

## Round 0.3 · accepted · Accept

My concerns and the reviewers' concerns have been adequately addressed. The scientific value of this revised article will push the field forward. The paper is ready to be considered for acceptance in this journal.

Reviewer 3 ·

Basic reporting

No comment.

Experimental design

No comment.

Validity of the findings

No comment.

Additional comments

No comment.